# Regeneration of Antifog Performance of Laser-Induced Copper-Based Micro-Nano Structured Surfaces by Rapid Thermal Treatment

**DOI:** 10.3390/nano14171415

**Published:** 2024-08-29

**Authors:** Huixing Zhang, Xinyi Xie, Xiaowen Qi, Chengling Liu, Chenrui Wang, Xiaolong Fang, Youfu Wang, Hongtao Cui, Ji Dong

**Affiliations:** 1School of Mechanical Engineering, Tianjin Sino-German University of Applied Sciences, Tianjin 300350, China; 2Department of Materials Science, School of Civil Engineering, Qingdao University of Technology, Qingdao 266520, China

**Keywords:** superhydrophilic, antifog, laser ablation, rapid thermal treatment

## Abstract

In this investigation, the laser marker ablation technique was employed on Cu-coated glass to fabricate micro-nanostructured antifog glass. The resulting surfaces exhibited a quasi-periodic micron hillock-hollow structure with dispersed nanoparticles distributed throughout, which played a role in the antifog property and superhydrophilicity. However, airborne organic pollutant deposition degraded the superhydrophilicity of ablated glass surfaces and, therefore, their antifog performance, which cannot be circumvented. Conventionally, furnace annealing for at least 1 h was used to decompose the organic pollutants and restore the superhydrophilicity, limiting the throughput and application scenario. Remarkably, the rapid regeneration of this property was achieved through either a 5 min rapid thermal treatment at 400 °C or a 1 s flame treatment. These are interventions that are hitherto unreported. Such short and simple treatment methods underscore the potential of laser-ablated glass for diverse practical applications.

## 1. Introduction

Fogging on automotive glass poses a significant challenge, often leading to hazardous situations, particularly during inclement weather conditions such as rain or cold weather. Similarly, fogging on medical endoscope lenses can block sight, potentially endangering lives [1,2]. Consequently, extensive research efforts have been directed towards developing effective antifog solutions across various domains, including optical instruments [3], sensors [4], medical lenses [5], and automotive glass [6].

Conventional methods for fog removal typically involve heating or introducing airflow; however, these approaches are both time and energy intensive [7,8,9]. Alternatively, researchers have explored the fabrication of superhydrophilic or superhydrophobic surfaces as promising strategies [6,10]. While superhydrophobic surfaces repel water droplets via rolling-off and bouncing-off effects [11,12], preventing fogging, they are prone to wetting over time due to droplet accumulation and growth within surface features [10], transitioning from a Cassie state to a Wenzel state [3]. To sustain superhydrophobicity, smaller nanostructures are preferred, although they may be susceptible to erosion from outdoor wind and sand. Moreover, the fabrication of superhydrophobic surfaces remains complex and technically challenging [13]. In contrast, superhydrophilic surfaces facilitate the rapid spreading of water, forming a continuous film that effectively eliminates fogging without compromising transparency [6,14]. Consequently, superhydrophilic surface preparation has emerged as a widely adopted approach for fog prevention [15,16,17].

SiO_2_- and TiO_2_-based coatings have been explored over the past few years, yet still suffer from insufficient durability and transmission loss [18,19,20,21,22,23,24]. Laser processing offers a flexible, non-contact, and controllable method for surface micro-nanostructuring without the need for chemical coatings or cleanroom facilities [25]. Direct laser writing on metals forms micro-nano structures on surfaces, which enhances surface roughness and generates a large amount of polar bonds on surfaces, leading to superhydrophilicity [26,27,28,29,30,31,32,33,34,35,36]. Such superhydrophilicity has been widely used in oil–water separation, antifogging, self-cleaning, water collection, and biomedical applications. Special lasers such as ultrashort lasers are required to directly process glass [37,38,39]. However, they may cause damage to the glass itself [40] and are expensive, featuring a low energy density, which leads to inefficient ablation and requires slow and costly multiple pulses [41,42]. The laser marker ablation of Al/V/Ni/Si/Ti-coated glass was tried in our previous papers, leading to the formation of micro-nano structures and their corresponding metal oxides on the glass surface, achieving a relatively durable antifog performance [43,44,45,46,47,48]. Cu is known to be corrosion resistant as well as antibacterial [48], and Cu oxide is photocatalytic [47,49]. The present study explores the laser marker ablation of Cu-coated glass. The adhesion of nanosized Cu films to glass substrates is critical for various technological applications, including microelectronics, where inadequate adhesion can lead to electrode delamination, ultimately compromising device reliability and functionality [50,51,52]. Further adhesion enhancement in our case may also be worth investigation. Meanwhile, the inevitable deposition of airborne organic pollutants on surfaces, followed by their adsorption, can compromise antifog performance by increasing water contact angles [53]. The adsorption process consists of chemical adsorption through esterification between airborne carboxyl and hydroxyl on the surface, and physical adsorption via hydrogen bonds between the airborne ester groups and the surface hydroxyl groups [54]. The dominant presence of non-polar C-C(H) bonds within the adsorbed organics degrades the superhydrophilicity, since high surface polarity typically results in strong hydrophilicity and vice versa [42,53,54,55]. Meanwhile, non-polar C-C(H) bonds have a low surface energy and tend to increase the surface contact angle according to Young’s theory [56,57], which is detrimental to the hydrophilicity of a surface. Thermal annealing at 250 °C and above for 1 h was reported to decompose the deposited organic pollutants and break the connection force between the surface and the organic pollutants, resulting in the recovery of the contact angle and the superhydrophilicity of the laser-ablated surfaces, which was the shortest attempt for restoring the superhydrophilicity [54,58]. Such time- and energy-consuming treatment would limit the throughput and the actual application scenario. To address this challenge efficiently, rapid thermal annealing and flash flame burn treatments are investigated for their potential to remove deposited organic pollutants and to regenerate antifog properties. The simplicity of flame treatment enabled the restoring process to be applied to many scenario such as automotive glass and even resin glasses.

## 2. Experiments

### 2.1. Materials and Methods

The target material utilized in this investigation was a high-purity 99.99% Cu target from Sino New Material, Zhengzhou, China. Soda-lime glass slides measuring 100 mm × 100 mm × 3 mm and 50 mm × 50 mm × 3 mm from Yaohua Glass Group, Qinhuangdao, China, served as the substrates. The substrates underwent a cleaning process involving a 10 min ultrasonic treatment in deionized water, anhydrous ethanol, and acetone to eliminate surface contaminants, followed by drying in an oven. Subsequently, the cleaned substrates were transferred to the sputtering chamber, where the background pressure was maintained below 1.0 × 10^−3^ Pa. Argon gas was the working gas with a flow rate of 30 sccm, while the sputtering pressure was maintained at 1.6 Pa. The film thickness (100 nm, 200 nm, 400 nm, and 600 nm) was varied for comparison.

The laser ablation of Cu-coated glass was performed using a pulsed laser marker (YLP-M30 from Qingdao Botai Laser, Qingdao, China) operating at a wavelength of 1064 nm, a pulse width of 100 nanoseconds, a repetition rate of 30,000 Hz, and an output power of 30 watts. The ablation procedure was conducted in a computer-controlled raster scanning mode, with a hatch distance of 0.03 mm. The scan rate was fixed at 1.0 m/s. Samples were labeled according to the thickness of the Cu coating. For instance, the sample designation “200 nm-Cu” indicates a sample processed with a 200 nm Cu coating and a 1.0 m/s laser scan rate.

Rapid thermal annealing was conducted in a furnace at 400 °C for 5 min, while flame treatment involved exposing the samples to the flame tip region from a flame gun for one second. This was carried out by moving the flame quickly across the entire surface of the sample, which was fixed on a sample rack. It took approximately one second to cover a 5 × 5 cm^2^ sized sample. A longer treatment may cause cracking due to thermal shock to the glass. The temperature of the flame tip reached approximately 1300 °C.

### 2.2. Characterization

A scanning electron microscope (SEM, Sigma300 by Carl Zeiss, Jena, Germany) equipped with an Energy-Dispersive Spectrometer (EDS) was adopted for morphology and elemental distribution analysis. Surface morphology and roughness characterization were performed using an Olympus OLS5100 laser scanning confocal microscope (LSCM, Tokyo, Japan). The three-dimensional (3D) surface structure at the nanometer level was acquired using an atomic force microscope (AFM), specifically the Bruker Dimension Icon from Ettlingen, Germany. The AFM measurement was taken in tapping mode, the tip radius was 8 nm, the resonance frequency range was 150 kHz, the scanning speed was 0.996 Hz, and the image resolution was 256 × 256 for the 1 μm × 1 μm range.

The surface chemical compositions of the samples were analyzed using X-ray photoelectron spectroscopy (XPS), utilizing the Thermo Scientific K-alpha instrument (Waltham, MA, USA). Contact angle measurements, serving as an indicator of surface wettability, were conducted using the goniometer DLDA23-01 from Zhuhai, China. For this measurement, 1 µL of water droplets was dispensed onto the sample surface using a micro-pipette. The droplet images captured by the camera were processed using the manufacturer developed software Drop Analyzer Interface, during which the contours of the droplets were fitted with ellipses. The tangent at the three-phase contact point was subsequently identified, enabling the calculation of the contact angle. The measurement was undertaken at room temperature (25 °C) and the resolution of the camera was 1440 × 1080. Furthermore, transmission measurements within the wavelength range of 400–1100 nm were carried out using the spectrophotometer Lambda 950 from Perkin Elmer (Waltham, MA, USA).

A boiling water vapor test was conducted to assess the antifogging performance of the samples. A hot water kettle was employed to boil water, and the samples were positioned at a distance of 40 mm above the kettle cover for a duration of 10 s each. The antifogging efficacy of every sample was observed, compared, and documented through photography under consistent background conditions. The background displayed the official website of the Greenwich Observatory indicating Beijing time.

## 3. Results and Discussion

Figure 1a–d present SEM images of ablated samples at a constant scanning speed of 1.0 m/s with various thicknesses of Cu coating. The images reveal the presence of irregularly shaped nanoparticles dispersed across the surfaces. These nanoparticles are believed to have formed as a result of plasma being generated during the process of material ablation and the subsequent recasting of the ablated material, as discussed in the previous literature [59]. Oxidation is inevitable due to the reaction between the highly active ablated Cu and the oxygen in the air. Highly irregular nanoparticles are observed in Figure 1a,d for a 100 nm and 600 nm Cu coating. For the 100 nm-Cu sample, an excessive ablation into the glass substrate occurred, resulting in the formation of numerous irregular particles due to the accumulated redeposition of neighboring particles. Conversely, excessively thick Cu coatings resulted in an abundance of irregular residuals left on the surface. Figure 1b,c show that a 200 nm and 400 nm Cu coating led to relatively dense and uniformly distributed nanoparticles. EDS mapping for Cu of a representative laser-ablated Cu-coated glass is shown in Figure 1e. It indicates that most of the Cu coating was ablated away, while the remaining Cu was widely dispersed on the sample surface. The micron-scale surface morphology and roughness of the treated samples was examined using SEM and LSCM, and the results are presented in Figure 1f,g and Figure 2 and Table 1. The images depict the presence of hillock and hollow structures characterized by shallow depressions surrounded by narrow protrusions. Similarly as in the SEM images, excessively thin and thick Cu led to non-uniformity, which hindered the water spreading to some extent. As shown in Table 1, 200 nm-Cu and 400 nm-Cu samples exhibited a relatively smooth surface. Figure 2f,g show the nano-scale morphologies of the 400 nm-Cu and 600 nm-Cu samples. Irregularly shaped nanoparticles were observed over the surface, which is consistent with the SEM results. The 400 nm-Cu sample exhibited a surface roughness Ra value of 6.60 nm, while the 600 nm-Cu sample exhibited a slightly higher surface roughness Ra value of 6.94 nm. Notably, the AFM results refers to nanoscale roughness within a highly localized 1 μm × 1 μm area, reflecting specific nanoscale features. In contrast, the roughness values presented in Table 1 refer to the micron-scale roughness that is measured using a confocal microscope over an approximately 1 mm^2^ area, providing an average roughness over a much larger surface. Confocal imaging offers a global perspective, while atomic force microscopy provides a more localized view. Nanoparticles are not uniformly distributed. Consequently, the roughness may vary across localized nanoscale regions. Therefore, roughness measurements from the confocal microscope may provide a more accurate overview of the surface’s roughness. According to the Wenzel model, an increase in surface roughness enhances the hydrophilicity of a surface [60], leading to a decrease in the contact angle.

Figure 3a,b show the XPS spectra of the 400 nm-Cu sample. It demonstrates that three main elements—C, O, and Cu—constituted these surfaces. The C1s peak, O1s peak, and Cu2p peak were around 284.8 eV, 531.5 eV, and 934.83 eV, respectively. The high-resolution Cu2p spectrum in Figure 3b shows that Cu and O were mainly bonded in the form of CuO. Both Cu and O tend to form polar bonds, which are instrumental to superhydrophilicity [61]. Only Cu was bonded in the oxidation state, and the surface plasmon effect may be negligible.

Figure 4 shows the transmission of laser-ablated samples with a fixed scan rate and varying Cu coating thicknesses. The 200 nm-Cu and 400 nm-Cu samples exhibited an improved transmission compared to the reference glass. Conversely, the 100 nm-Cu and 600 nm-Cu samples experienced a decrease in transmission compared to the reference glass. The 400 nm-Cu coating sample demonstrated the optimal transmission, showing a 1.5% broadband enhancement in transmission over the wavelength range of 400–1100 nm. This enhancement can be attributed to the densely and uniformly packed nanoparticle structure observed in Figure 1, which may be partly explained by the Fresnel equation, as follows:$$R={\left(\frac{n_{1}-n_{2}}{n_{1}+n_{2}}\right)}^{2}$$
where *n*_1_ and *n*_2_ are the refractive indices of the two media through which the light propagates. The reflection is determined by the refractive index difference between the two media. A negligible difference results in minimal reflection, and zero refractive index difference results in zero reflection. According to the effective medium theory [62], the surface with dispersed nanostructures behaves as a multi-media layer with a minimal refractive index gradient, as the nanostructures are much smaller than the wavelength of the incident light. Consequently, the reflection is reduced when the light propagates across the interface between the substrate and the surface layer, as it encounters minimal refractive index changes. Similarly, the reflection is reduced when the light propagates across the interface between the air and the surface layer, as if no interface exists between them. The water film coverage right after an antifog test reduced the transmission slightly because of the diffraction and reflection loss due to the water film. This slight reduction did not affect the vision at all. Both excessively thin and thick Cu coatings led to the formation of large irregular micro-nano structures, resulting in transmission losses due to either additional scattering or parasitic absorption.

Figure 5a,b demonstrate the antifog images and contact angle characterization of a specially fabricated sample, which consists of half treated 400 nm-Cu sample and half control glass. The contact angle was only measured on ablated surfaces. The treated part showcased a remarkable antifog performance and superhydrophilicity in comparison with the reference glass, which is attributed to the micro-nano structure and surface chemical bonds. After 22 months of storage in the laboratory, the treated part lost its antifog performance. A 1 s flame treatment restored the antifog performance of the sample, reducing the contact angle from 67° to 0°, which was rarely reported in the field. Wiping with a wet cloth did not affect the antifog performance, but did increase the contact angle slightly to 3°. Figure 5e–h show the antifog images before and after a 5 min rapid annealing treatment at 400 °C for a sample consisting of half treated 200 nm-Cu sample and half reference glass. Figure 5e,g illustrate a substantial degradation in the antifog effect of the treated portion after 6 months of storage in the laboratory. However, the antifogging performance of the sample was successfully restored after a 5 min 400 °C treatment, with the text behind the glass becoming clearly visible during the antifog test. Figure 5f,h demonstrate that the contact angle decrease from 48.5° to 5.8° after a 5 min 400 °C treatment.

Figure 6 shows the XPS spectra of the 400 nm-Cu sample after and before the 1 s flame exposure. Figure 6a shows that the flame-treated surface still mainly consisted of C, O, and Cu. The deconvolution of the C1s peak is illustrated in Figure 6b,c. Figure 6b shows that the C1s peak was resolved into three peaks at 284.8 eV, 285.85 eV, and 288.99 eV, corresponding to C-C, C-OR, and COOR, respectively. C-C was non-polar and adverse to superhydrophilicity. On the post-flame treatment surface, the C-C bonds accounted for only 58.87% of the carbon content, whereas on the pre-flame-treated surface, they accounted for 88.45% of the carbon content. The flame treatment burned off a larger amount of non-polar flammable organic pollutants containing C-C bonds on the surface than polar organic pollutants. This reduction in the content of C-C bonds indicates a decrease in non-polar components. It is well established that a high surface polarity typically results in a strong hydrophilicity [42]. This explains why the samples exhibited an increased hydrophilicity following a 1 s flame treatment.

Figure 7 presents the XPS spectra for another 400 nm-Cu sample subjected to different processing treatments. The amount of organic pollutants adsorbed on the surface varied among the samples due to differing air exposure durations, leading to distinct elemental compositions compared to those shown in Figure 6. Specifically, the overall carbon content on the surface decreased from 61.25% to 50.4%, and further to 40.58% after rapid annealing at 400 °C and flame treatment, respectively. Similarly, the relative concentration of C-C(H) bonds within the surface carbon content was reduced from 80.79% to 73.51% after rapid annealing, and to 72.46% following flame treatment. This reduction in both the overall carbon content and the relative abundance of C-C bonds is attributed to the thermal decomposition during rapid annealing and the combined effects of thermal decomposition and burning during flame treatment. The partial recovery of superhydrophilicity observed after rapid annealing at 400 °C suggests that this treatment only partially decomposes the adsorbed organics.

In contrast with Figure 1f,g, Figure 8 shows an SEM image of a representative sample before and after flame treatment. The surface morphology remained unchanged during rapid annealing, as demonstrated in Figure 1. Figure 8 presents more wrinkled regions after flame treatment, suggesting interface roughening, as the flame temperature was well above the softening point of glass. This subtle roughening may enhance the adhesion between the glass and the Cu-based surface layer, which was consistent with previous findings [50,51,52]. Figure 8 also indicates that the nanostructures on the hillock-hollow formations were not uniformly dispersed. Nanoparticles were primarily concentrated on the hillocks, while they were scattered and sparsely distributed in the hollows. The hollows represent overlapping ablation regions, resulting in minimal residual material or redeposition. This further verifies the roughness difference between the AFM and confocal microscopy results.

## 4. Conclusions

Hierarchical micro-nano structures featuring a micron-scale hillock-hollow architecture with widely dispersed nanoparticles atop the micron structures were fabricated on glass substrates via the laser marker ablation of Cu-coated glass to achieve superhydrophilicity. The Cu-based nanoparticles were predominantly concentrated on the hillocks, while they were sparsely distributed within the hollows. The study investigated the influence of Cu film thickness on the generated surface of the micro-nanostructures and its consequent effects on transmission, hydrophilicity, and antifog performance. Both excessively thin and thick Cu coatings were associated with drawbacks. Overly thin Cu coatings caused over-ablation of both the Cu film and the underlying glass, resulting in irregular micro-nano structures, which caused scattering loss and hindered water spreading. Excessively thick Cu coatings caused non-uniformity and an excess of residuals, also negatively affecting transmission and hydrophilicity. The optimal Cu thickness was determined to be 400 nm. The densely and uniformly distributed nanoparticles on the optimally processed glass surface minimized the refractive index change across the surface–substrate interface, thereby reducing the reflection. Consequently, the 400 nm-Cu sample demonstrated an excellent antifog performance and transmission improvement by 1.5% in the 400–1100 nm wavelength range compared to untreated glass. Remarkably, rapid thermal treatments at either 400 °C for 5 min or exposure to a 1 s flame reinstated the superhydrophilicity and antifog properties of the ablated surface, which has not been reported before. The flame treatment effectively removed adsorbed organic contaminants through both thermal decomposition and burning, surpassing the efficacy of rapid annealing at 400 °C, which only partially eliminated the organic pollutants through decomposition, leading to the incomplete recovery of superhydrophilicity.

## Figures and Tables

**Figure 1 nanomaterials-14-01415-f001:**
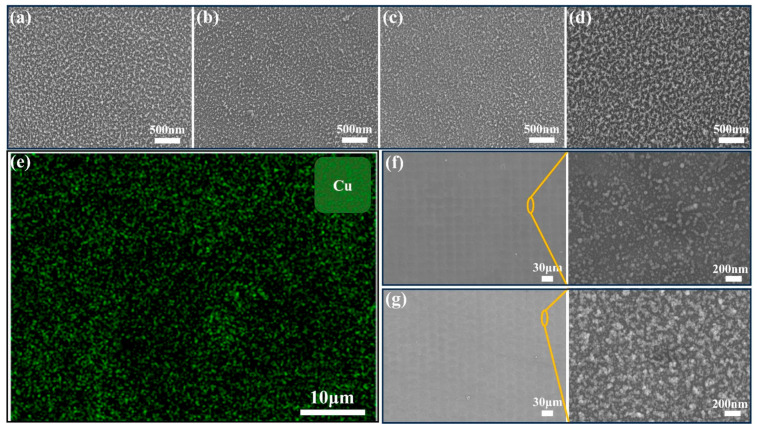
SEM images of laser-ablated glass with a scanning speed of 1.0 m/s and various Cu coating thickness. (**a**) 100 nm; (**b**) 200 nm; (**c**) 400 nm; (**d**) 600 nm; (**e**) EDS mapping for Cu element of a representative laser-ablated glass; SEM images of the 400 nm-Cu sample before and after rapid thermal annealing at 400 °C. (**f**) Before annealing; (**g**) after annealing.

**Figure 2 nanomaterials-14-01415-f002:**
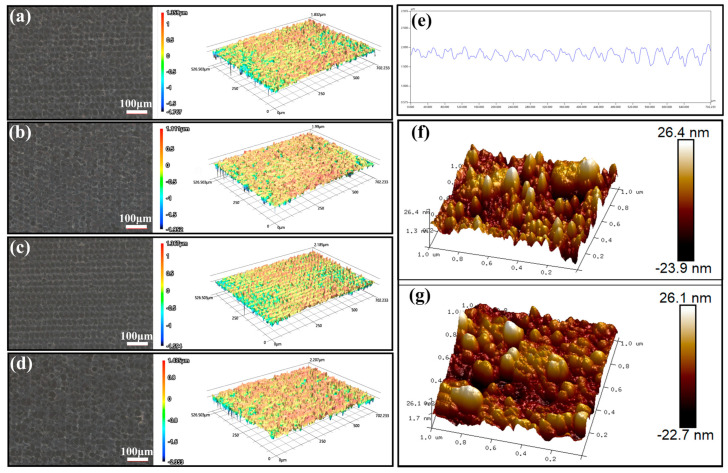
Confocal images of laser-ablated glass with a scanning speed of 1.0 m/s and various Cu coating thickness. (**a**) 100 nm; (**b**) 200 nm; (**c**) 400 nm; (**d**) 600 nm; (**e**) a typical cross-sectional profile image of a line in (**c**). AFM images of laser ablated samples. (**f**) 400 nm-Cu; (**g**) 600 nm-Cu.

**Figure 3 nanomaterials-14-01415-f003:**
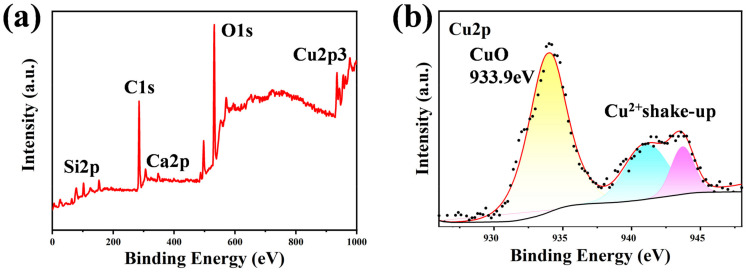
XPS spectra of the treated 400 nm-Cu sample. (**a**) Treated glass surface; (**b**) Cu2p region. The dots were raw data and the lines were fitted.

**Figure 4 nanomaterials-14-01415-f004:**
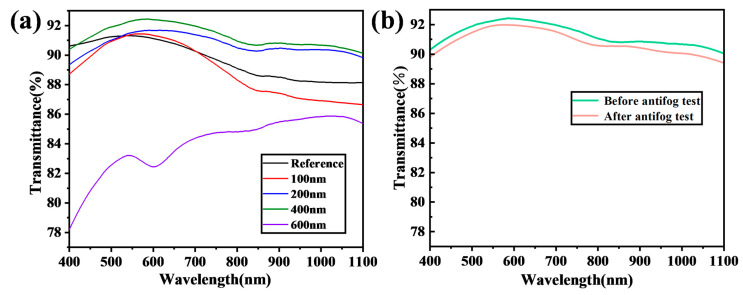
(**a**) Transmission of laser-ablated Cu-coated glass with various thicknesses (100 nm, 200 nm, 400 nm, and 600 nm) at a laser scanning rate of 1.0 m/s. (**b**) Transmission of ablated glass processed with a 400 nm Cu coating before and immediately after an antifog test.

**Figure 5 nanomaterials-14-01415-f005:**
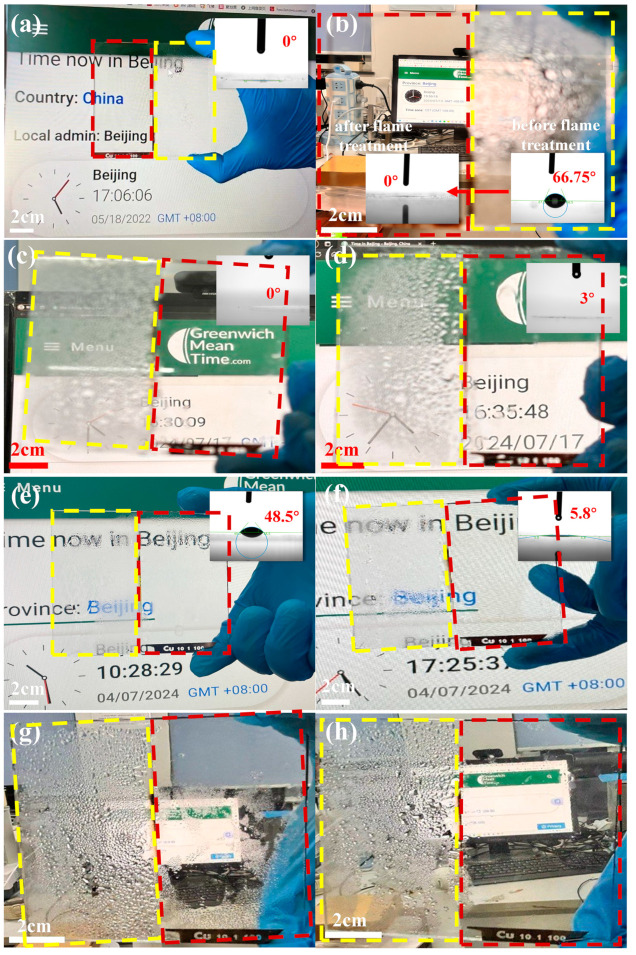
Antifog images and wettability characterization of a special sample consisting of half treated 400 nm-Cu sample and half control glass. (**a**) Freshly ablated sample; (**b**) regeneration of antifog effect after 22 month storage in the laboratory via 1 s flame treatment. Antifog images of the sample in (**b**) before and after wiping with a wet cloth: (**c**) before wiping and (**d**) after wiping. Antifog images and wettability characterization of another sample consisting of half treated 200 nm-Cu sample and half control glass, before and after a 5 min rapid annealing treatment at 400 °C. (**e**) Pre-annealing treatment after being stored in the lab for 6 months; (**f**) post-annealing treatment. (**g**,**h**) Far-field antifog images of samples in (**e**,**f**) before and after the rapid thermal annealing at 400 °C. The reference part was yellow framed and the treated part was red framed.

**Figure 6 nanomaterials-14-01415-f006:**
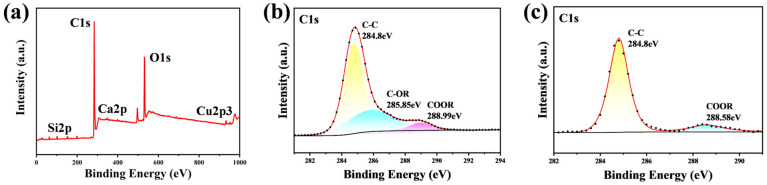
XPS spectra of the 400 nm-Cu sample. (**a**) Full spectrum (after flame treatment); (**b**) C1s region in (**a**); (**c**) C1s region in Figure 3a.

**Figure 7 nanomaterials-14-01415-f007:**
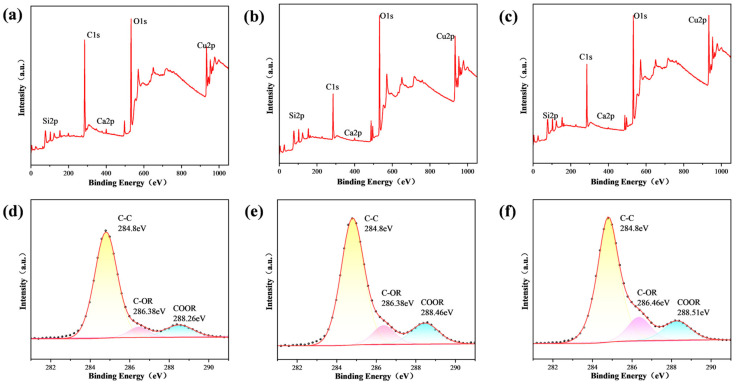
XPS spectra of a 400 nm-Cu sample subjected to different processing treatments. (**a**) Full spectrum after storage in the laboratory for approximately one year; (**b**) full spectrum of the sample in (**a**) after undergoing flame treatment; (**c**) full spectrum of the sample in (**a**) after undergoing rapid thermal annealing; (**d**) C1s region for the sample in (**a**); (**e**) C1s region for the sample in (**b**); (**f**) C1s region for the sample in (**c**).

**Figure 8 nanomaterials-14-01415-f008:**
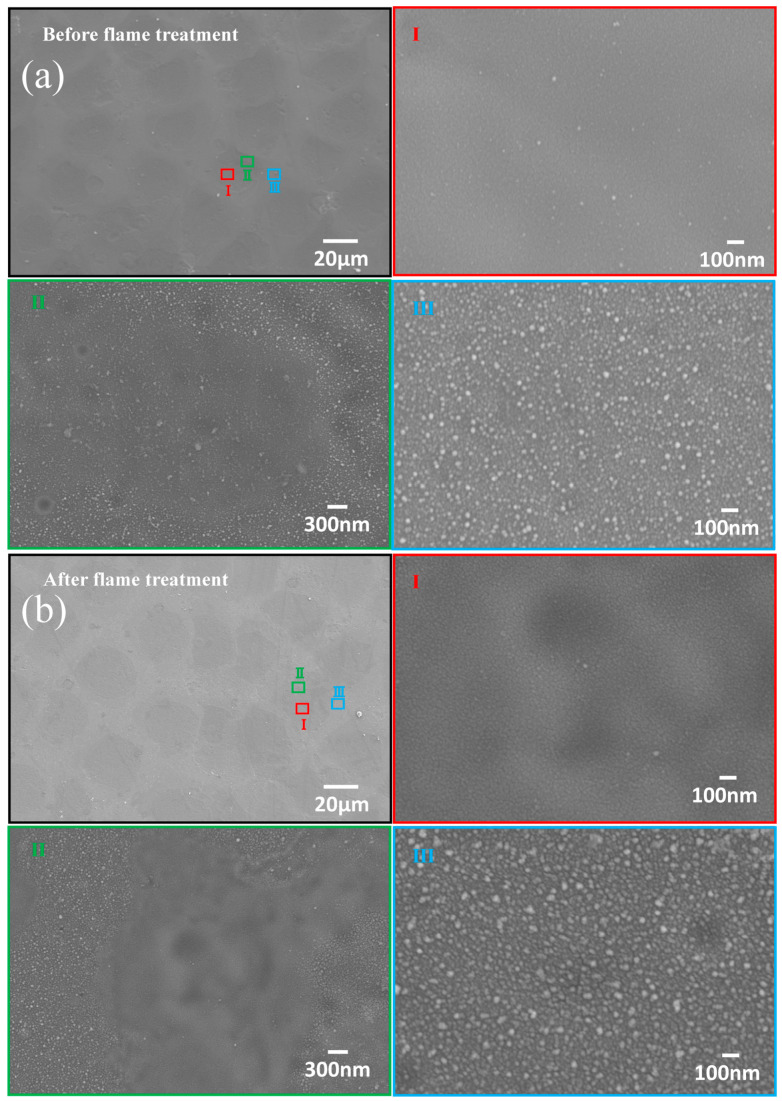
SEM images of a representative sample before and after flame treatment. (**a**) Before flame treatment; (**b**) after flame treatment.

**Table 1 nanomaterials-14-01415-t001:** The surface roughness (Ra) of the samples processed at various Cu coating thicknesses.

Sample Name	100 nm-Cu	200 nm-Cu	400 nm-Cu	600 nm-Cu
Surface roughness (μm)	0.472	0.447	0.437	0.604

## Data Availability

Data is contained within the article.

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
