# Peer review of "Regeneration of Antifog Performance of Laser-Induced Copper-Based Micro-Nano Structured Surfaces by Rapid Thermal Treatment"

_nanomaterials, 2024, doi:10.3390/nano14171415_

Round 1

Reviewer 1 Report

Comments and Suggestions for Authors

This paper describes an experimental research to fabricate superhydrophilic surface by laser ablation of metal-coated glass plate. The objective is clear and attractive to many readers, the methods seem well described, and the results are soundly presented. I would like to recommend the manuscript to be published in Nanomaterials. However, I found several issues to be considered or modified before publication.

(1)   About the surface roughness: In the text, it describes “Sample 400 nm-Cu exhibited a surface roughness Ra value of 6.60 nm, (L. 139)” for example, which seem contradictory to Table 1, which gives values of 0.4~0.6 mm.

(2)   Relating to (1), characters in Fig. 1 is too small to discern, especially in (i). Some enlargement is preferable.

(3)   Figure 3: I cannot understand why the transmittance of 200 and 400 nm coating surface is larger than the reference glass sample. More detailed specification is needed for the “reference”. Maybe laser-ablated surface with no Cu coating?

(4)   In Conclusions: Could you more elaborate on “hierarchical” micro-structure?  Probably this refers to the fabricated surface covered with the nanoparticles, but the relation between the nanoparticles and larger scale structures is not so clearly described.

Reviewer 2 Report

Comments and Suggestions for Authors

The paper looks good and shows promising results for improved antifog performance of the nanostructures. I would recommend publication of this paper in nanomaterials after the authors incorporate my comments in the revised version of the manuscript.

Reviewer 3 Report

Comments and Suggestions for Authors

The paper iz on Regeneration of Antifog performance of laser induced copper based micro-nano structured suface by rapid thermal treatment

The comments and suggestions are listed bellow.

Introduction:

Please describe and add reference for Cu coated glass.

Please describe and add reference for efect of C-C bonds on hydrophilicity.

Experimental:

Is it possible to estimate the temperature of a flame gun?

Is it suggested to reorganize Fig. 1 in that way to make images more visible

Results and discussion:

Fig 2a. Please indicate the part of Cu coating that was ablated away.

Page 5., Line 166. It is hard to belive that sample with  Cu nano coating has higher transparency than the reference samples just due to minimal refractive index gradient. Please discuss.

Fig. 4. Please indicate the control glass.

Please explain the difference between thermal treatment and exposure to flame.

Comments on the Quality of English Language

Minor editing.

Round 2

Reviewer 3 Report

Comments and Suggestions for Authors

Unfortunately the revision is not complete.

The comments that are not responded are listed bellow.

Introduction:

Please describe and add reference for Cu coated glass. Not responded.

That means to describe what has been done in the field of Cu coated glass for application where fogging is critical (not microelectronics). 

Experimental:

Is it possible to estimate the temperature of a flame gun? Responded.

The temperature of flame is estimated to be 1300 C. Bearing in mind that glass possesses low thermal shock resistance it would be essential to explain how cracking due to thermal shock was avoided during sample exposure to flame. Is high temperature of flame is expected to cause Cu melting or oxidation?

Results and discussion:

Page 5., Line 166.(the first version) It is hard to believe that sample with  Cu nano coating has higher transparency than the reference samples just due to minimal refractive index gradient. Please discuss. Not responded.

The passage about the refractive index does not explain the increase of transparency of coated sample. Even if the refractive indices are the same there is still a question: why the transparency of coated glass is higher than that of reference glass

Comments on the Quality of English Language

English is good.
